# Conventional Therapeutics in BPDCN Patients—Do They Still Have a Place in the Era of Targeted Therapies?

**DOI:** 10.3390/cancers14153767

**Published:** 2022-08-02

**Authors:** Margaux Poussard, Fanny Angelot-Delettre, Eric Deconinck

**Affiliations:** 1RIGHT Interactions Greffon-Hôte Tumeur/Ingénierie Cellulaire et Génique, UMR1098, EFS BFC, INSERM, University Bourgogne Franche-Comté, F-25000 Besançon, France; margaux.poussard@efs.sante.fr (M.P.); fanny.delettre@efs.sante.fr (F.A.-D.); 2Etablissement Français du Sang Bourgogne Franche-Comté, Laboratoire d’Immuno-Hématologie, F-25000 Besançon, France; 3Service d’Hématologie, CHRU Besançon, F-25000 Besançon, France

**Keywords:** BPDCN, conventional therapeutics, targeted therapies, chemotherapies, allogeneic stem cell transplantation

## Abstract

**Simple Summary:**

Since its recognition in 2016 as a distinct entity among acute myeloid leukemia (AML), no consensus on treatment has been established for managing blastic dendritic cell neoplasm (BPDCN). Patients seem sensitive to standard chemotherapies, but relapses and resistance development often occur. To date, only allogeneic stem cell transplantation presents better results with extended overall survival. New targeting therapies appear regularly, offering therapeutic options. Here, we discuss the therapies currently available and the sequence of treatments that may be proposed to patients. We tried to determine the place of standard chemotherapy and allogeneic transplantation among these new targeted treatments for the BPDCN population.

**Abstract:**

No benchmark treatment exists for blastic plasmacytoid dendritic cell neoplasm (BPDCN). Since the malignancy is chemo-sensitive, chemotherapy followed by hematopoietic stem cell transplantation remains an effective treatment. However, relapses frequently occur with the development of resistance. New options arising with the development of therapies targeting signaling pathways and epigenetic dysregulation have shown promising results. In this review, we focus on conventional therapies used to treat BPDCN and the novel therapeutic approaches that guide us toward the future management of BPDCN.

## 1. Introduction

Plasmacytoid dendritic cells represent <0.5% of nucleated blood cells in healthy individuals. They play a central role in immune response and interact closely with the monocytic lineage; however, they are also involved in some cancers. A pDC expansion can lead to an immunosuppressive and protumoral environment, especially in hematological diseases. Blastic plasmacytoid dendritic cell neoplasm (BPDCN) arises directly from pDC over-expansion. BPDCN is a rare hemopathy classified among acute myeloid leukemia (AML) since 2008 by the World Health Organization (WHO). As a specific entity since 2016, it represents <1% of AML. We estimated that BPDCN represents 0.44% of new hematologic neoplasms annually, that is, an incidence of 0.04 new patients per 100,000 people, equating to approximately 700 new cases annually in the USA and 1000 new cases annually in Europe [1]. BPDCN mainly affects elderly patients with a clear male predominance (male/female ratio of 3:1), even if some pediatric cases were reported [2,3,4]. Some authors suggest a possible bimodal distribution with two peaks of incidence <20 years old and >60 years old. In 90% of cases, the first clinical sign is a cutaneous involvement that rapidly spreads to peripheral blood, bone marrow, and lymph nodes, leading to the patient’s death in one or two years. BPDCN shows poor prognostic features with a median overall survival between 9 and 23 months. Given its rarity, aggressiveness, and diagnosis difficulty, it is challenging to establish a treatment consensus for BPDCN to date. Even if a good overall response is observed with ALL or AML regimens, rapid relapses with drug resistance occur in most cases. Only young and fit patients receiving aggressive polychemotherapy followed by hematopoietic stem cell transplantation (HSCT) in first complete remission (CR) obtained long-term remissions with a median survival of 30 months. However, older patients, representing the majority of affected patients, are often unfit and unable to receive these heavy treatments; therefore, other strategies are being developed. Targeted therapies such as hypomethylating agents, BCL-2 inhibitors, or NF-kB inhibitors were investigated and showed promising results. Treatments targeting the central nervous system (CNS) were also questioned. CNS involvement has been highlighted several times as a potential sanctuary site inaccessible to standard therapies. It accounts for up to 20% of relapses if prophylactic treatment is not included in the frontline therapeutics. More recently, the development of immunotherapies has experienced a great surge, especially those targeting CD123, constantly overexpressed by BPDCN cells.

In this review, we describe conventional therapeutics used for patient care in real life and all the targeted molecules or regimens currently in evaluation, except for CD123-based targeted therapies described elsewhere in this Special Issue.

## 2. Conventional Therapies and New Therapeutic Options for BPDCN

### 2.1. ALL Regimen vs. AML and Lymphoma Regimens

The most common treatment for BPDCN is induction therapy based on regimens used for acute leukemia (both AML and ALL) or non-Hodgkin’s lymphoma (NHL). NHL regimens generally comprise CHOP (cyclophosphamide, doxorubicin, vincristine, prednisone) or CHOP-like regimens. ALL regimens comprise hyper-CVAD (hyperfractionated cyclophosphamide, vincristine, doxorubicin, and dexamethasone alternating with high-dose methotrexate and cytarabine) regimens or high doses of methotrexate combined with asparaginase. AML regimens usually comprise cytarabine combined with daunorubicin or idarubicin. In a retrospective study, Taylor et al. compared the use of these different regimens as a first-line treatment for BPDCN [5]. Their study was based on 59 patients identified in three different centers in the United States. Patients treated with first-line lymphoid-type regimens showed improved progression-free survival (PFS) compared to those treated with myeloid regimens (2-year PFS; 40% vs. 11%, respectively; *p* = 0.075). They also highlighted the impact of the regimen intensity used. Indeed, intensive regimens have led to a significantly increased PFS compared with non-intensive ones (2-year PFS; 45% vs. 11% respectively; *p* = 0.034). Other retrospective studies also confirmed the positive outcome experienced by patients treated with lymphoid-type regimens [6,7,8,9,10]. These studies have shown the necessity of consolidation to obtain prolonged survival. Laribi et al. showed that patients treated with lymphoid-type regimens followed by HSCT consolidation presented higher CR rates and lower relapse rates (ALL-type regimen: 94% achieved CR and 13% relapsed; NHL-type regimens: 100% achieved CR and 33% relapsed) than those treated with AML-type regimens followed by HSCT consolidation (88% achieved CR and 58% relapsed) [7]. Thus, they confirmed that lymphoid-type treatments produce better outcomes; however, superior effects with a prolonged OS are only observed when consolidation is performed for eligible patients with HSCT. Based on collections established by the French BPDCN network, Garnache-Ottou’s group identified treatments received by patients in a series of 89 well-documented cases [11]. They also demonstrated that leukemia-like regimens exhibited the highest response rate compared to less intense approaches with a median remission duration of 47 months in the AML-like (anthracyclines associated with cytarabine as in “5 1 7” AML treatments)/ALL-like (multi-drug associations as in ALL treatments)/AspaMTX (high-dose methotrexate with asparaginase) group compared with 7 months (*p* = 5.038) in the CHOP-like (classical regimen used in the treatment of non-Hodgkin lymphomas and combining cyclophosphamide, doxorubicin, vincristine, and prednisone)/NOS group (all other drugs alone or in combination). Finally, in a recent publication on their website, the German Society of Hematology and Oncology published recommendations highlighting the use of leukemia regimens for eligible patients with BPDCN [12].

Several cases have reported using multi-drug therapy combining methotrexate, L-asparaginase, and dexamethasone in this pathology [13]. This combination is well-tolerated, with a response rate of approximately 70% for patients whose age and comorbidities allow for allo-HSCT [13,14,15]. Our team has previously demonstrated high efficacy of idarubicin on BPDCN primary cells with <1% of viable cells remaining after 18 h of treatment [16]. Based on our previous results, we proposed the first prospective phase II trial for patients with BPDCN to evaluate the effectiveness of three cycles of methotrexate, L-asparaginase, idarubicin, and dexamethasone for eligible patients, followed by allo-HSCT when CR is achieved (Figure 1). This phase II study has been recruiting patients since January 2019 and has already included 20 patients with preliminary results for publishing soon (NCT03599960 Combination Chemotherapy in Patients with Newly Diagnosed BPDCN (LpDessai)). Preliminary data confirmed that the disease is a chemo-sensitive entity with a significant rate of first remissions and some adverse events with the need to monitor frail patients closely. The first security analysis emphasizes the remaining importance of adverse events in the frail population of BPDCN patients (unpublished data).

### 2.2. Allo-HSCT

For many years, allo-HSCT has been the standard consolidation treatment for BPDCN patients who are fit for this therapy and achieve CR1. The results depend on the intensity of the conditioning regimen (myeloablative conditioning or reduced-intensity regimen), type of treatment, stage of disease, initial presentation, and age of patients. In a meta-analysis involving 128 patients, Kharfan-Dabaja et al. showed a pooled OS of 67% for patients allografted at CR1 versus 7% for patients who received an allo-HSCT beyond CR1 [17]. In the same way, a retrospective study based on 86 patients identified in the French BPDCN network highlighted the efficacy of allo-HSCT in CR1 with no regard to previous treatment received. Among the 30 patients who received allo-HSCT, only 10 (33%) relapsed after a median time of 12 months, whereas all patients receiving auto-HSCT relapsed after 2, 4, 7, and 17 months (4/4 (100%)) [11]. A recent study by Brüggen et al. confirmed the superiority of allo-HSCT over any type of chemotherapeutic regimen [8]. In a clinical study involving 398 patients published in 2020, Laribi et al. showed that chemotherapy followed by allo-HSCT in CR1, and to a lesser extent auto-HSCT, was associated with a significantly better outcome [7]. In the absence of a randomized controlled trial comparing allo-HSCT with conventional chemotherapy or novel therapies in patients with BPDCN, and according to all published data, the optimal approach to achieve a long-term remission is allograft in the first CR regardless of the regimen used to obtain this remission [9]. However, whether novel targeted therapies can be used without allotransplants as a consolidation remains a pending question.

### 2.3. Auto-HSCT

Few studies or limited data of case reports are available for an autologous-HSCT (auto-HSCT) evaluation. The vast majority of studies have shown the superiority of allo-HSCT, especially in CR1. However, Aoki et al. showed in a study of 25 patients receiving auto-HSCT in CR1 a 4-year PFS and OS of 73% and 82%, respectively, which are superior to patients receiving allo-HSCT (PFS: 60% and OS: 69%) [18]. Nevertheless, these positive results must be nuanced. Numerous other studies do not share the same conclusion, as highlighted by Garnache-Ottou et al., whose four patients treated with auto-HSCT relapsed compared to only 10/30 of patients treated with allo-HSCT [11]. To date, it is difficult to recommend it as a standard approach for BPDCN due to the limited number of auto-HSCT cases. This issue can probably be discussed in patients without initial marrow involvement (for instance, isolated cutaneous presentation) who also obtained a CR on all involved sites.

### 2.4. BCL-2 Inhibition

Targeted therapies have been more developed and evaluated in recent years since they are less aggressive and present promising results. Among them, Montero et al. performed an in vitro and in vivo proof of concept for the efficacy of venetoclax in BPDCN models overexpressing BCL-2 protein and relying on it to survive [19]. Numerous case reports [20,21] and clinical trials have also reported good results with fast regression of skin nodules and disease control even after several lines of treatment. However, relapses occur often, and studies have been implemented to evaluate them in combination with other agents. One of the most studied combinations is with 5-azacytidine, already approved (since 2018) for treating newly diagnosed AML for patients >75 years old or with comorbidities and ineligible for intensive induction chemotherapy [22,23,24,25,26]. Thus, a combination of hypomethylating agents is currently under investigation in two different clinical trials against BPDCN (NCT03113643 and NCT03404193). Its association with low-intensity chemotherapy was also presented by Di Nardo et al. as a viable salvage option even in relapsed or refractory (R/R) patients, particularly those presenting a *RUNX1* and/or *IDH1/2* mutation [27]. In the same way, Cherry et al. compared venetoclax plus 5-azacytidine to an induction chemotherapy (IC) regimen in AML management. They showed that the OS for venetoclax associated with 5-azacytidine was more favorable than the IC regimen in a propensity-matched cohort of patients with equivalent baseline factors. They also identified variables that can guide treatment decisions, such as RUNX1 mutation, which favors venetoclax plus 5-azacytidine over the IC regimen [28].

These results showed that targeting BCL-2 protein remains an interesting strategy but leads to good short-term results when used alone. In order to prolong their efficacy, their association with other molecules is required. A combination with cell therapy may be considered since several studies have shown the positive impact of venetoclax on immune cells, especially T cells [29,30,31,32].

### 2.5. Epigenetic Dysregulation Targeting

To better describe and understand BPDCN, Sapienza et al. studied whole-exome sequencing (WES) and its genetic characteristics. Several epigenetic modifiers were identified as bearing mutations, such as *ASXL1* and *TET2* [33,34]. Therefore, we studied hypomethylating agents for BPDCN patient treatment, which showed promising results. The most commonly tested was the 5-azacytidine, approved by the FDA for the treatment of myelodysplastic syndrome in 2004 [35]. Although it showed good results as a monotherapy, more studies have reported its use in association with other treatments. Its capacity to modulate gene expression by hypomethylation may be used to overcome resistance issues and thus enhance other treatment efficacy. A resistance to tagraxofusp, a CD123-targeted therapy, was described by Lane et al. in a BPDCN model [36]. This resistance works through the *DPH1* expression loss required for the efficacy of tagraxofusp. Azacytidine restored *DPH1* expression and thus the cell’s sensitivity to tagraxofusp. Its evaluation in a clinical trial with tagraxofusp is currently in progress (NCT03113643). It was also evaluated in association with venetoclax in two clinical trials and has been widely used in case reports, as this combination shows good results (NCT04216524 and NCT03113643).

### 2.6. Multiple Myeloma-Based Regimens

#### 2.6.1. NF-κB Inhibition

Previous alternative treatment options targeted the NF-κB pathway. It is an interesting approach since BPDCN exhibits constitutive activation of the NF-κB pathway [37]. Moreover, NF-κB inhibition in BPDCN cell lines using either an experimental specific inhibitor JSH23 or the clinical drug bortezomib interferes in vitro with leukemic cell proliferation and survival [38]. Our team confirmed that bortezomib inhibits the phosphorylation of the RelA NF-κB subunit efficiently into two BPDCN cell lines in vitro and in vivo. We demonstrated that bortezomib could be associated with other drugs used in different chemotherapy regimens (i.e., a histone deacetylase vorinostat (SAHA) and statins) to improve its impact on BDPCN cell death [39]. In 2019, Marmouset et al. also described a promising combination of lenalidomide/bortezomib and dexamethasone with two complete responses and one complete remission [40] after five cycles of this chemotherapy. These results justify using bortezomib in combination with other chemotherapies for treating BPDCN patients.

#### 2.6.2. CD38 Targeting

A recent case report showed an encouraging clinical response in an elderly BPDCN patient treated with daratumumab-based therapy after prior azacytidine-venetoclax. Daratumumab is an anti-CD38 monoclonal human immunoglobulin G1 antibody that induces cell death and has important immunomodulatory activities in myeloma, which may also play a role in BPDCN [41].

### 2.7. CD123-Targeted Therapies

#### 2.7.1. Tagraxofusp

In the area of targeted therapies, target choice is a key point. For BPDCN, one largely described target was highly overexpressed by 100% of the patients. This target of choice was the alpha subunit of the IL3 receptor, called CD123. Among all the therapies developed to target it, SL-401 is the most described. Based on several clinical trials, it received US FDA approval for untreated or relapsed or refractory (R/R) BPDCN patients in December 2018 [42]. Afterward, it received EU approval in January 2021. The first clinical trial was led by Frankel et al., who presented an ORR of 78% (7/9 patients) comprising five CRs and two PRs [43]. Some toxicities were noted but presented as manageable. Pemmaraju et al. led a prospective and multi-institutional study to confirm these first results on both frontline (FL) and R/R BPDCN patients. They observed 90% of ORR in the FL group, with 72% of patients presenting a CR or CRc. The OV at 2 years was 52%, and 45% of the patients were bridged to HSCT. In the R/R group, 67% of ORR was observed with an OS of 8.5 months. For the entire trial, 20% of patients presented CLS, causing two deaths [44].

Currently, SL-401 is in several clinical trials on CD123 hemopathies in combination with azacytidine and venetoclax to enhance its efficacy [45] (NCT03113643 and NCT04216524).

#### 2.7.2. Other Therapies Targeting CD123

IMGN632 (Immunogen, Waltham, MA, USA) is a humanized monoclonal antibody targeting CD123, fused to a potent DNA alkylating agent. It was shown to be active in vitro and in vivo against BPDCN models, and studies suggested that normal hematopoietic stem cells should be preserved at their low expression of CD123 [46]. IMGN632 is currently being tested alone on untreated or R/R BPDCN patients; the first results showed an ORR of 30% (7/23) and CRc of 22% (NCT03386513) [47]. No grade 3 or higher adverse events were noted (except for one patient). The study is still ongoing in Europe and the US. Based on this preliminary evidence, IMGN632 received a Breakthrough Therapy Designation in October 2020 for the treatment of BPDCN.

The surface expression of CD123 makes it a good target for immunotherapy, such as CAR-T cells (chimerix antigen receptor). Thus, since the beginning of the 2010s, numerous teams have worked on developing CD123-directed CAR-T cells and demonstrated their efficacy in AML and BPDCN. More academic teams and private companies are working on developing such products [48,49,50]. Almost 30 trials are ongoing, evaluating CD123 CAR-T cells, mainly in AML, but some include BPDCN patients. No trials have ended, but some high toxicities are noticeable, such as cytokine release syndrome and capillary leak syndrome (due to the targeting of normal cells expressing CD123 at a low level) (NCT03203369). Thus, some works and trials are now adding a safety part in the CAR construct to limit toxicity (NCT02159495 and NCT04109482) [51,52].

### 2.8. Future Perspectives

#### 2.8.1. BET Inhibition

New epigenetic regulatory factors have recently emerged as new targets for cancer treatment. Among these, the BET proteins (bromodomain and extraterminal domain proteins), which recognize acetylated histones to regulate gene activity, are interesting. BET proteins have deregulated expression and/or activity in cancers and represent major players in maintaining transformed phenotypes and therapeutic resistance. The recent introduction of small molecules inhibiting BET proteins offers the possibility of abrogating their oncogenic functions. Recent work has highlighted the interest of BET inhibitors (BETis) in BPDCN [53]. Ceribelli et al. have shown that these molecules induce apoptosis of BPDCN cells via a TCF4 transcription factor (E-box transcription factor). TCF4 controls an array of target genes in BPDCN, such as *MYC*. In addition, Emadali et al. recently showed that the haploinsufficiency of NR3C1, a gene encoding a glucocorticoid receptor found in 28% of BPDCN patients, was linked to BPDCN to BETi sensitivity [54]. In a recent review, Cheng Chen et al. highlighted the efficacy of JQ1 on CAL-1 (BPDCN cell line) both in vitro and in vivo [55]. Other studies have also shown that BETi induces cellular apoptosis and suppresses tumor growth in BPDCN xenograft models by downregulating *TCF4*. BETi is a promising therapeutic alternative with nearly 20 BETi in preclinical trials.

#### 2.8.2. LXR Agonists

Ceroi et al. showed that treatment with a liver X receptor (LXR) agonist decreases leukemic cell infiltration and BPDCN-induced cytopenia while increasing survival in a BPDCN-induced mouse xenograft model [38]. These results suggest that LXR-induced STAT5 and NF-κB inhibitions may be involved in both the inhibition of cell proliferation and BPDCN cell death, demonstrating a new therapeutic approach where cholesterol homeostasis is modified in BPDCN and normalized by treatment with LXR agonists.

#### 2.8.3. Cladribine

A cladribine-based regimen was reported in 2020 for a patient with relapse after receiving three cycles of chemotherapy, mostly AML regimens [56]. The patient underwent inhibitor therapy targeting *RUNX1* and intensive chemotherapies but no clinical benefits. The patient received a CLAAG regimen profile: cladribine 5 mg/m^2^ and cytarabine 1 mg/m,15-day continuous infusion, G-CSF priming, liposomal adriamycin 20 mg/m^2^, and 2-day continuous infusion. He received two cycles of the CLAAG treatment profile. After the second cycle, a complete response for morphology was observed. The authors suggested that the patient has benefited clinically from the cladribine regimen, a synthetic purine nucleoside analog medicine.

### 2.9. CNS Involvement

Even if some studies have highlighted interest in CNS prophylaxis treatment, neurological implications have not been deeply investigated. In a prospective study, Martin–Martin et al. showed that all but one of the six patients presenting occult CNS involvement at diagnosis who received intrathecal (IT) treatment survived [57]. By contrast, patients with occult CNS involvement at relapse/progression died even after receiving IT treatment. Their results were confirmed in a retrospective study of 23 BPDCN patients, suggesting CNS could be a persistent blast-cell sanctuary in BPDCN patients. Pagano et al. showed a high rate of neuromeningeal involvement in 43 patients (16%), both at diagnosis and relapse. However, among relapsed patients, those presenting CNS relapse were patients that did not receive prophylactic IT therapy. These results suggested that a systematic preventive IT chemotherapy was indicated during first-line treatment to prevent CNS relapse and thus lead to a better outcome for patients. More recently, Sapienza et al. discovered new neural features by comparing 19 BPDCN patients to 4 normal pDC samples [58]. They observed 51 deregulated miRNAs in BPDCN samples that significantly influenced neurogenesis. The neurogenic process was significantly enriched in BPDCN samples compared with pDCs. They showed overexpression of *NLGN4X* and *EDN3* in BPDCN samples, but only *NLGN4X* was specific to BPDCN (as this overexpression was also present when comparing the samples to AML samples). No neural cells were found in the microenvironment of BPDCN patient samples, but tumor cells showed expressions of *DCX* (11/15) and *UCHL-1* (15/15), which are neural markers involved in neurogenesis. The BPDCN tumor thus seems to retain some neural properties. Collectively, these results highlighted the neural-oriented transcriptional program of a BPDCN tumor with the expression of well-known neural genes (*EDN3*, *NLGGN4X*, *UCHL-1*, and *DCX*) and the activation of neural receptor genes (Ach receptors). They may all be involved in the cross-talk between BPDCN cells and CNS, leading to CNS relapse and progression of the disease. These new and promising results opened a new axis of investigation regarding treatments proposed to patients.

## 3. Discussion

In 2022, some consensus is now well-established concerning the clinical aspects and the diagnostic features of BPDCN. Nevertheless, the treatment is still a real challenge with no consensus in the scientific literature. Only one drug, tagraxofusp, was recently approved in the USA and Europe as a specific first-line treatment in BPDCN; however, accessibility and reimbursement of the drug are not assured in all countries. Tagraxofusp is the first CD123-approved targeted therapy; nevertheless, it cannot be considered a reference treatment for BPDCN. In this section, we discuss different treatment options for this rare form of leukemia (Table 1).

Although BPDCN is an aggressive disease, positive responses are easily observed in first-line treatment regardless of which chemotherapy regimen was used. Steroids or monotherapy with methotrexate sometimes offer a clinical CR. However, relapses occur rapidly, and the tumor progresses fast. The major problem is not to obtain the first CR but to maintain this CR. The current therapeutic strategy to treat BPDCN patients is to trigger a lasting remission using induction agents that bring them to the subsequent allograft. The toxicity of the induction regimen is a key element for the success of further allografts, and less toxic induction regimens are preferred. The general status and fitness of the patients must also be considered and are key elements in this perspective of systematic allograft requirements. The ALL regimen appears to trigger better responses in fit patients than AML or lymphoma regimens, even if some discrepancies exist between the two largest retrospective series [7,11]. However, without consolidation with HSCT, one specific regimen’s superior effects over another are not maintained. In terms of HSCT, numerous studies have shown the superiority of allo-HSCT over auto-HSCT. In the largest retrospective series [7], the role of auto-HSCT offers some survival advantages in older patients not eligible for allo-HSCT. For other authors, the place of auto-HSCT must be considered for patients without any initial bone marrow or blood involvement. For now, allo-HSCT remains the standard of care for the consolidation of fit patients in the first CR.

On the other hand, BPDCN patients are often too old and/or weak to benefit from allograft and are thus eligible for new therapies even in a frontline setting. Several options are available today with a wide range of low-dose chemotherapies or targeted therapies. Targeted therapies can be chosen based on immunophenotyping of blast cells or genomic analyses but also on current understanding of the physiopathological pathways of the disease. However, long-term remissions are difficult to maintain with those regimen types, and prolonged efficacy is rarely observed. Some molecules described previously in this article are of particular interest. Bortezomib has shown solid preclinical data, and some case reports or small series outlines show encouraging clinical results. Venetoclax, a BCL-2 inhibitor, is also suggested in some recent case reports. Daratumumab was also recently reported to bring BPDCN patients to CR. Some other molecules cited above have shown anecdotal results. Combining these new molecules, particularly bortezomib, lenalidomide, and steroids, in the same manner as multiple myeloma may be interesting. These combinations may also include CD123-targeted therapies. Very high CR rates are now reported with hyper-CVAD plus tagraxofusp, for instance. Alternatively, a less toxic approach may involve a combination of azacytidine, venetoclax, and tagraxofusp.

The major chance of response for all these treatments, including new molecules and targeted therapies, is observed in frontline treatment. For relapsing patients, the chance of obtaining a second remission is <30%; even if the patient is transplanted in the second remission, the chance of prolonged survival is <10%.

Special attention must be drawn to the prophylactic meningeal treatment associated with standard chemotherapy and new therapeutic approaches. The sole validated treatment is the intrathecal administration of chemotherapy (generally cytarabine and/or methotrexate) during the induction phase.

## 4. Conclusions

No consensus treatment currently exists for the standard care of BPDCN patients (Figure 2). There are no scientific arguments for choosing between ALL, AML, or lymphoma-like regimens, and the choice must be adapted to the anticipated toxicity profile of the patient. Prophylactic intrathecal chemotherapy must be added to the induction regimen for all patients. All eligible patients must be transplanted in the first remission, preferably with an allogenic donor; in some cases, autologous transplantation is a good alternative. Determining the place of new molecules and targeted therapies is still challenging, even with encouraging results. The question of using a combination of chemotherapy, new molecules, and targeted treatments is still open and must be explored to avoid the systematic hematopoietic transplantation requirement for most patients. Similar to other hematological malignancies, we may also consider the possibility of prolonged maintenance therapies.

## Figures and Tables

**Figure 1 cancers-14-03767-f001:**
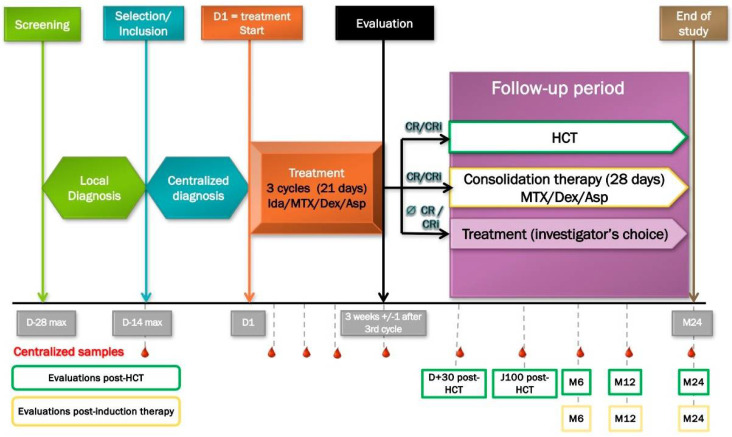
LpDessai clinical trial (NCT03599960) design. D, day; Ida, idarubicin; MTX, methotrexate; Dex, dexamethasone; Asp, asparaginase; CR, complete remission; Cri, complete remission with incomplete hematopoietic recovery; HCT, hematopoietic stem cell transplantation; M, month; wk, week.

**Figure 2 cancers-14-03767-f002:**
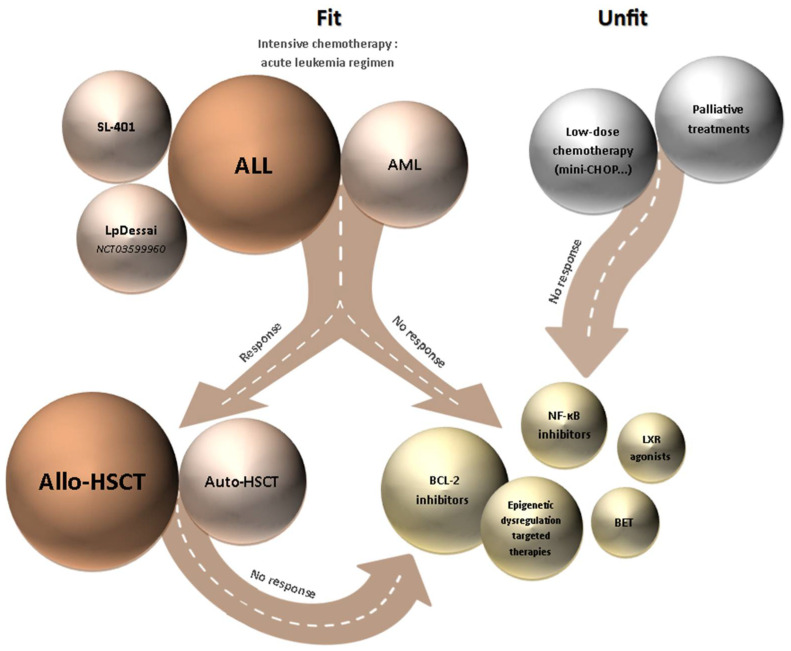
Diagram of current BPDCN management. The size of the circles represents the efficacy and frequency use of the treatment. The color of the circle represents the patient group the treatment is recommended to (fit (orange), unfit (gray), or after several failures (gold)). ALL, acute lymphoid leukemia; AML, acute myeloid leukemia; Allo-HSCT, allogenic hematopoietic stem cell transplantation; Auto-HSCT, autologous hematopoietic stem cell transplantation; BET, bromodomain and extraterminal domain; LXR, liver X receptor.

**Table 1 cancers-14-03767-t001:** Summary of studies and ongoing clinical trials on BPDCN patients evaluating conventional and targeted therapies.

Conventional and Approved Therapies						
Treatments Methods	n	CR n (%)	Relapse n (%)	OS (Mon)	PFS (%)	References
ALL-type	35	n.r.	n.r.	n.r.	at 2 years: 40	Taylor et al. [5]
AML-type	9	n.r.	n.r.	n.r.	at 2 years: 11	
ALL-type + allo-HSCT	33	31 (94)	4 (13)	n.r.	n.r.	Laribi et al. [7]
NHL-type + allo-HSCT	12	12 (100)	4 (33)	n.r.	n.r.	
AML-type + allo-HSCT	16	14 (88)	2 (58)	n.r.	n.r.	
NHL/ALL/AML-type + auto-HSCT	16	n.r.	5 (31)	n.r.	n.r.	
AML-type	19	13 (68.4)	4 (28.5)	18	n.r.	Garnache-Ottou et al. [11]
ALL-type	15	15 (78.9)	5 (33.3)	15	n.r.	
Aspa-MTX	16	12 (75)	4 (33.3)	15	n.r.	
CHOP-type	16	6 (37.5)	4 (66.7)	11	n.r.	
allo-HSCT	30	n.r.	10 (33)	49	n.r.	
auto-HSCT	4	n.r.	4 (100)		n.r.	
NHL-type	10	5 (50)	n.r.	n.r.	n.r.	Yun et al. [6]
ALL-type	11	10 (91)	n.r.	n.r.	n.r.	
AML-type	1	n.a.	n.r.	n.r.	n.r.	
SL-401	12	6 (50)	n.r.	n.r.	n.r.	
allo-HSCT (in CR1)	110	n.r.	n.r.	pooled: 67%	pooled: 53	Kharfan-Dajaba et al. [17]
auto-HSCT (in CR1)	19	n.r.	n.r.	pooled: 7%	pooled: 7	
allo-HSCT (in CR1)		n.r.	n.r.	at 4 years: 69%	at 4 years: 60	Aoki et al. [18]
auto-HSCT (in CR1)	25	n.r.	n.r.	at 4 years: 82%	at 4 years: 73	
SL-401	9	5 (55%) [2 (22%) PRs]	n.r.	n.r.	n.r.	Frankel et al. [43]
SL-401	32 FL	90% of ORR	n.r.	at 2 years: 52%	n.r.	Pemmaraju et al. [44]
	15 R/R	67% of ORR	n.r.	8.5	n.r.	
**Under evaluation targeted therapies**						
**Treatment methods**	**n**	**Type of study**	**Results**		**Status**	**References**
Venetoclax	2	Case report	PR at 4 weeks		/	Montero et al. [19]
	1	Case report	CR at 5 months, no new cutaneous lesions at 10 months		/	Grushchak et al. [20]
	/	Phase 1	n.a.		Recruiting	NCT03113643
	/	Phase 2	n.a.		Recruiting	NCT03404193
5-azacytidine	/	Phase 1	n.a.		Recruiting	NCT03113643
	/	Phase 2	n.a.		Recruiting	NCT04216524
Lenalidomide/bortezomib/dexamethasone	3	Case report	2 CR and 1 clinical remission		/	Marmouset et al. [40]
IMGN632	/	Phase 1/2	n.a.		Recruiting	NCT03386513
CAR-T cells	/	Phase 1	n.a.		Recruiting	NCT04318678, NCT02159495
	/	Phase 1/2	n.a.		Recruiting	NCT04109482

BPDCN, blastic dendritic cell neoplasm; ALL, acute lymphoblastic leukemia; AML, acute myeloid leukemia; NHL, non-Hodgkin’s lymphoma; Aspa-MTX, asparaginase-methotrexate; CHOP, cyclophosphamide, doxorubicin, vincristine, prednisone; allo-HSCT, allogeneic hematopoietic stem cell transplantation; auto-HSCT, autologous hematopoietic stem cell transplantation; CAR-T, T cells expressing a chimeric antigen receptor; CR, complete remission; OS, overall survival; PFS, progression-free survival; Mon, month; n.r., not reported; n.a., not available.

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
