# Peer review of "Conventional Therapeutics in BPDCN Patients—Do They Still Have a Place in the Era of Targeted Therapies?"

_cancers, 2022, doi:10.3390/cancers14153767_

Round 1
Reviewer 1 Report
In their review, Poussard et al address a relevant topic, which is well-justified based on the open issue on whether targeted therapies achieve similar efficacy / disaease control than conventional options.
There are some issues that preclude publication it its current form.
DATA AND COMPARISONS
Although a great effort is made to report available data on conventional and targeted therapies, it misses two major things:
1) A short paragraph on the comparison with Tagraxofusp or other CD123 targeting approaches.
2) A summary table in which all major (efficacy) data are aligned in a visually easy to navigate way, including conventional options, targeted options, including CD123 targeting strategies.
This will tremendously help in interpreting the Figure 2.
In figure 2, the circle sizes are explained, but not their color-codes.
STYLE AND LANGUAGE:
Meanings of some of the phrases are often lost in translation. At too many places there are “non-English” expressions. Moreover, the text is filled with typos and grammar errors.
Some few examples are:
- A phase II study is being recruited since (it actively recruits)
- Preliminary data confirmed that the disaese (disease)
- is a 121 chemosensitive entity with a significant rate of first remissions (COMMA) but also some adverse 122 events with the neccessity to BE closely monitored
- …whether novels (NOVEL) directed
- Epigentic dysregluation targeting (EPIGENETIC) // Targeting of epigenetic dysregulation
- An example of non-English: “Even if some studies highlighted the interest ???? of CNS prophylaxis treatment, so far 274 neurological implication ?????? was ???? not deeply investigated.
Author Response
We took into account all the comments made regarding datas and style.
1) A paragraph describing tagraxofusp and other CD123 targeting approachs currently under evaluation was added to the manuscript.
2) A summary table highlighting the efficacy of the different regimens was also added to the manuscript.
In general, style and language were revised.
Reviewer 2 Report
This is a comprehensive review of the therapeutic management of the rare condition of blastic dendritic plasmacytoid cell neoplasms. As mentioned by the authors, most results in the literature are based on small series, which makes the exercise difficult and recommendations hazardous. However, this manuscript will likely be found helpful.
General comments
The LpDessai, especially since it is still ongoing with no communicable results, should be mentioned in the ALL/AML chemotherapy section, not as a specific paragraph.
Cladribin does not deserve a specific heading either and could go as 2.8.3, thus leaving the defintely different question of CNS involvement as the last part of the document.
It would be interesting to explain that tagraxofusp is an original combination including diphtheria toxin, not a standard monoclonal antibody. In fact, CD123 targets could probably deserve a specific paragraph.
The discussion is highly redundant and should be shortened.
More importantly, the document suffers from poor english grammar and many typos and requires consequent editing to make it readable.
Minor comments
Genes should be italicized
Drugs should be named in lower font (no capital letter) as per DCI/INN recommendations.
Author Response
We took into account all the comments made regarding datas and style.
- The paragraph describing the LpDessai was merged with the one talking about ALL/AML regimen.
- Cladarabin section was moved to "Future perspectives" section as recommended.
- A specific paragraph describing CD123 targeted therapies was added to the manuscript.
- The discussion was revised and shortened.
Regarding the style and language, the manuscript was revised and modified to improve english grammar and typos (genes italicized and drugs named in lower font).
Reviewer 3 Report
In the current review, the authors summarize the conventional chemotherapies for Blastic dendritic cell neoplasm (BPDCN) except for CD123 targeted therapies. The manuscript is well written, and the readers can understand the contents easily. Here are my comments that would improve the already great manuscript.
1. It is well recognized that CD123 targeted therapies in BPDCN have become Gamechangers for BPDCN. From the current manuscript, it may be unclear about the association between CD123 targeted therapies and the conventional chemotherapies though I understand that the current review article focuses on the other therapies than CD123 targeted ones. For example, it is unclear how CD123 targeted therapies are treated in the Figure 2. It would be great if the authors could describe the points.
2. (Lines 205-207) In terms of the genomic mutations of BPDCN, a recent study (PMID: 34615655) also summarized them comprehensively using more samples than those in the study that the authors cited as “ref. 33”
Author Response
We took into account all the comments made regarding datas.
- A specific paragraph describing CD123 targeted therapies was added to the manuscript and integrated in Figure 2.
- Paragraph dealing with epigenetic dysregulation was completed with the reference you suggested (PMID: 34615655).
Round 2
Reviewer 1 Report
The authors made an effort to address the issues raised. However, as a reviewer it is nearly impossible to track / discern the old wording from the new, because the old writing is neither erased nor stroken. So, spaces and new hybrid words make proper reading impssible.
The English really needs a thorough overhaul; i.e. non-English expressions (e.g. "In view to better describe" ... ???) or spellings e.g. Chimerix Antigen Receptor (chimeric)
The Figure legend still has no indicative call-out of the colors and their meaning.
Author Response
An english editing was performed by MDPI and the changes are indicated in the file.
Figure legend was completed to better describe colors meaning.
Reviewer 2 Report
it is much improved and now acceptable.
Author Response
An english editing was performed by MDPI and changes are indicated in the file.